# Incidence of Intoxications in the Emergency Department of Galati Hospital with Examples of Cardiovascular Effects of MDMA Intoxication

**DOI:** 10.3390/diagnostics13050940

**Published:** 2023-03-01

**Authors:** Liliana Dragomir, Virginia Marina, Mihaela Anghele, Aurelian-Dumitrache Anghele, Cosmina Alina Moscu

**Affiliations:** 1Clinical-Medical Department, Faculty of Medicine and Pharmacy, “Dunărea de Jos” University of Galati, 800201 Galati, Romania; 2Medical Department of Occupational Health, Faculty of Medicine and Pharmacy, “Dunărea de Jos” University of Galati, 35 Str. Domneasca Galati, 800201 Galati, Romania; 3Department of General Surgery, Faculty of Medicine and Pharmacy, “Dunărea de Jos” University, 800201 Galati, Romania; 4Emergency Department of Hospital of Galati, 800201 Galati, Romania

**Keywords:** methylen-dioxy-methamphetamine, intoxication, serotonin drug, amphetamine

## Abstract

MDMA (3,4-methylenedioxymethamphetamine; commonly referred to as “Molly” or “ecstasy”) is a synthetic compound, structurally and pharmacologically similar to both amphetamines and mescaline. MDMA differs somewhat from traditional amphetamines in that it is not structurally similar to serotonin. Cocaine is rare and cannabis is consumed less frequently than in Western Europe. Heroin is the drug of choice for the poor in Bucharest, Romania’s capital of two million people, and alcoholism is common in villages where more than a third of the population lives in poverty. By far, the most popular drugs are Legal Highs (Romanians call them “ethnobotanics”). All of these drugs have significant effects on cardiovascular function that contribute significantly to adverse events. Most adverse cardiac events occur in young adults and are potentially reversible. Poisoning among patients aged 17 years and over was commonly seen in the Emergency Departments of a large tertiary hospital in the city centre, accounting for 3.2% of all patients. In a third of the poisonings, more than one substance was used. Intoxication with ethnobotanicals was the most frequently observed, followed by use of drugs from the amphetamine group. The majority of patients presenting to the Emergency Department were male. Therefore, this study suggests further research on hazardous alcohol consumption and drug abuse.

## 1. Introduction

The average drug overdose mortality rate in Europe is estimated at 17 deaths per million people aged 15–64 years [1] and accounts for about 3.5% of all deaths in adult men under 40 years [2].

National mortality rates vary considerably and are influenced by factors such as pattern of drug use, particularly injecting drug use, characteristics of the drug-using population and reporting practices. Among heroin users, overdose occurs frequently and the main associated risk factors are age at onset of dependence, severity of dependence and concomitant use of other substances, particularly alcohol and benzodiazepines [3,4,5,6].

Romania is one of the poorest member states of the European Union and there is a total lack of awareness about addiction treatment and a high level of stigma towards alcoholics and drug addicts. Few people in Romania are aware that addiction is a disease and that treatment programmes are available [1,2].

Recreational drug use has reached epidemic levels. Forty-five million people in the European Union have used cannabis at some time, with proportionately more young people using it [7]. The use of high-risk drugs such as cocaine and heroin are increasing, with around 1.5 million users in the European Union. Drug use is frequently associated with complications, including an increased risk of premature death [7,8].

MDMA (3,4-methylenedioxymethamphetamine, commonly referred to as “Molly” or “ecstasy”) is a synthetic compound, structurally and pharmacologically similar to both amphetamines and mescaline.

Ecstasy (MDMA) and related drugs are amphetamine derivatives that also have some of the pharmacological properties of mescaline. Ecstasy (MDMA) is the popular name for a substance chemically identified as N-methyl-3,4-methylenedioxyamphetamine or 3,4-methylenedioxy-methamphetamine. A closely related compound is N-ethyl-3,4-methylenedioxyamphetamine. The name ‘ecstasy’ is used somewhat indiscriminately. In earlier years, the name was also given to 3,4-methylenedioxyamphetamine (MDA). The three compounds are closely similar in their chemistry and biological effects. Ecstasy differs from amphetamine and methamphetamine, but resembles the structure of the hallucinogenic material mescaline. As a result, the pharmacological effects of MDMA are a mixture of those of amphetamines and mescaline.

This difference probably explains the increased release of serotonin and consequent inhibition of serotonin reuptake [9,10,11].

The literature confirms the existence of minor adverse reactions such as neuropsychomotor agitation, bruxism, nausea, diaphoresis, ataxia, visual changes, cardiac changes (tachycardia and hypertension). These effects are usually self-limiting and will subside spontaneously within a few hours [12].

Cocaine is rare and cannabis is consumed less frequently than in Western Europe. Heroin is the drug of choice for the poor in Bucharest, Romania’s capital of two million people, and alcoholism is common in villages where more than a third of the population lives in poverty. By far, the most popular drugs are Legal Highs (Romanians call them “ethnobotanics”). Unfortunately, Legal Highs are also the most dangerous drugs on the market, and doctors do not know what they contain.

All of these drugs have significant effects on cardiovascular function that contribute significantly to adverse events. Most adverse cardiac events occur in young adults and are potentially reversible. The key to diagnosis is a high index of suspicion, especially when unexplained or unusual cardiovascular problems occur in association with central nervous system dysfunction, along with awareness of the pathophysiological effects of the drugs [13].

Alcohol and drug intoxications are a common problem in the Emergency Department. In the literature, approximately 1–5% of all Emergency Department visits were due to single or multiple intoxication [1,2].

Intoxications range from alcohol, drugs (pharmaceutical, non-pharmaceutical or illicit drug use/drugs of abuse) to carbon monoxide (CO) and chemicals. Globally, alcohol intoxication is the most prevalent intoxication among patients presenting to the ED [2,3,4,5,6,14,15,16].

Over time, changes in the number of specific types of poisoning were observed in other countries [17,18,19,20,21,22,23].

There is an increase in visits due to alcohol consumption in Emergency Departments in Romania [17,18]. There is also an increasing trend in presentations due to drug use, although not all studies support these results [18,19,20]. Among pharmaceuticals, an increase in ED visits for benzodiazepines and opioids was reported up to 200% in Romania [21,21,22,23].

Alcohol-impaired patients are considered a burden on healthcare around the world. This burden includes social and economic aspects as well as the burden imposed on hospitals [24,25,26,27,28]. The increase in Emergency Department visits and hospital admissions due to intoxication has been described in the literature. Subsequent use of healthcare resources is part of a growing burden on our healthcare system [27].

However, no recent data are available on health care consumption due to poisoning in Romanian Emergency Departments.

Therefore, the current study aimed to describe the occurrence and characteristics of intoxications presented to the Emergency Department in the Galati Emergency Clinical Hospital, Galati, Romania.

## 2. Materials and Methods

The study population consisted of patients over 17 years of age, who required medical care due to a single or multiple poisoning during 2014–2016, in the Emergency Department of the Clinical County Hospital Galati, Galati, Romania.

### 2.1. Inclusion and Exclusion Criteria

The poisonings included drugs (pharmaceutical, non-pharmaceutical or drugs), carbon monoxide (CO) and chemicals. For this study, an intoxication was reported as present if the attending physician described the intoxication in the patient’s medical record.

Inclusion criteria were: patients aged ≥17 years who presented to the Emergency Department between 1 January 2014 and 31 December 2016, who either presented with a poisoning as a primary or secondary reason. A secondary reason was defined as presenting to the Emergency Department for a reason other than intoxication, but upon review, it became clear that medical care was needed for a present intoxication.

A number of variables were taken from the electronic medical records. Variables included patient characteristics (gender, age, intentional versus unintentional exposure), intoxication characteristics (type, co-intoxications), therapy variables, day and time of presentation, admission to the ward or Intensive Care Unit (ICU).

### 2.2. Characteristics of Eligible Studies

The study was conducted in accordance with the World Medical Association Declaration of Helsinki, using a protocol approved by the local Bioethics Committee, and obtaining approval from the hospital administration (Protocol code: 5257/02.03.2021).

A total of 609 patients met the inclusion criteria for this study, representing 0.21% of all presentations in the Emergency Department.

## 3. Results

### 3.1. Characteristics of the Studied Batch

During 2014–2016, a total of 258,160 patients (age ≥ 17 years) were consulted in the Emergency Department (Table 1).

The average age of patients presenting to the Emergency Department was 51 years for substance intoxicated patients (range 17–82), and 26 years for patients who had used ethnobotanicals. Of all patients presenting with intoxication, 82.9% were male, with the age range 17–60 years predominating (Figure 1).

Most of the intoxicated patients presented to the emergency room on weekends. Almost half of the patients came during the night.

Twenty-five percent of all poisonings were admitted to hospital, of which 15% were admitted to the ICU. The 66% remaining patients were discharged home, 4%, to a psychiatric, addiction or homeless centre, 1% to the police station and 4%. left without medical advice. Of all patients admitted, 24% were admitted to the Internal Medicine ward and 22% to the neurology ward; 15% of patients admitted were admitted to the ICU, representing 4% of the total group of poisonings. Almost a quarter (24%) of all admitted patients were transferred to another hospital; more than half of these were admitted to the ICU, i.e., 55% (Table 2).

In 75% of all intoxications, some form of diagnosis was performed, mainly general blood samples (54%) and electrocardiograms (41%). Radiological examinations were performed in all patients; predominantly brain computer tomograph (26% of all intoxications).

Gastric lavage was performed in 1% of patients and charcoal was administered in 3% of all patients. Drugs other than an antidote were administered in 38% of patients, including benzodiazepines which were administered in 4.3%. Three percent of patients were intubated.

### 3.2. Characteristics of Intoxications

Ethnobotanical drugs of abuse caused 66.9% of all intoxications. The mean age of patients with a drug intoxication was 31 years (range: 16–69; SD 10.9) and 67% of patients were male. The distribution of different types of intoxication according to the age variable is shown in Figure 2.

The most commonly observed drug intoxications were ethnobotanicals for 407 patients, followed by heroin use (70 patients), and other drugs were MDMA (17 patients), cocaine (24 patients), amphetamines (28 patients), cannabis (10 patients) and ecstasy (35 patients). We also note that men were predominant in this category as well (Figure 3).

Chemicals were involved in 26% of the poisonings; mainly unknown chemicals (66 patients), chlorine (17 patients) and detergents (7 patients). Additionally, poisonings with paint or thinner (-vapors) (6 patients), antifreeze (4 patients), carbon monoxide (3 patients), herbicides, pesticides or organophosphates (10 patients), descaler (4 patients), sodium hydroxide (caustic soda) (4 patients), formaldehyde (1 patient), copper sulphate (1 patient), plant fertiliser (1 patient) and rodenticides (5 patients) (Table 3).

### 3.3. Cases Report—Study on the Different Outcomes of Two Patients Admitted to the ED for Acute MDMA Intoxication

Next, the evolution of two patients was compared. Both patients were male and 32 years old. From the clinical observation sheets, we found that the two patients presented on the same day to the Emergency Department of the Emergency Hospital of Galati (on 23 April 2021). The patients were brought by ambulances from the same village.

Due to the fact that the two patients were brought by the ambulance service at the same time to the Emergency Department, this work was carried out in a comparative manner, constantly presenting the management of the patients, by exposing their evolutions simultaneously, in order to detect the risk factors that finally led to the discharge of both subjects (with a slow favourable evolution). Taking into account the condition of the patients at the time of admission to the Emergency Department (Figure 4), we continued the presentation of the main manoeuvres performed in the Emergency Department.

The patient L.I.

Medical procedures:-Two peripheral venous lines;-Intra-osseous access;-Mechanical ventilation + oro-tracheal intubation without induction;-Continuous monitoring (blood pressure, SaO_2_ (Oxygen Saturation), the ventricular allura, the ejection fraction), EKG-Nasogastric tube mounting;-Urinary bladder catheter mounting (500 mL of normal urine);-The cardiological examination: the Cardio-respiratory arrest ≥ with external cardiac massage;

Cardiopulmonary arrest resuscitated with resumption of sinus rhythm; the inferior-posterior myocardial infarction with right ventricular damage;

Ultrasound-chord: Ejection fraction = 40%, contractility disturbance localized to the inferior wall, posterior lateral, no evidence of fluid in the pericardium;

The patient was referred to the Intensive Care Unit, continuously monitored. The patient received the prescribed treatment;

PCR SARS-CoV-2 Test: Negative;

The cranial brain computer tomography 23 April 2021: no recent injuries;

The abdominal ultrasound no recent injuries.

The patient C.I.

Medical procedures: Peripheral venous access. Continuous monitoring (blood pressure, SaO_2_, ventricular allura, ejection fraction, EKG). The urinary bladder catheterization (hyper chromic urine 1200 mL);

The cardiology examination: sinus tachycardia of unspecified etiology). The patient’s general condition was very bad, conscious but hardly responsive to the stimuli (verbal and painful);

Cardiac ultrasound: the left ventricle undiluted, with normal global systolic function; no fluid deceleration in the pericardium. The patient was referred to the Intensive Care Unit where he is continuously monitored;

The patient received the prescribed treatment;

PCR SARS-CoV-2 Test: Negative;

The cranial brain computer tomography 23 April 2021: no recent injuries;

The abdominal ultrasound: intestinal stasis; failure to visualize the pancreas; the liver homogeneous; the porta vena = 12 mm; the gallbladder without calculi. the spleen = 119 mm; the normal kidneys; no free intraperitoneal fluid.

The distinct evolution of the two patients, analysed on the basis of the information obtained from the observation records, during the first 3 days post-admission was observed. The following can be observed:First patient L.I.: the patient presented leukocytosis in remission during this period, with a gradual decrease in PLT (up to a value of 105). Glycemic values were maintained above the threshold of 150 mg/dL, with a maximum value of 183 mg/dL. Normalization of liver indicators values (with remission of hepatocytolysis syndrome). At the same time, the CRP value was over 30 × normal (Table 4).The second patient, C.I., presented evolving leukopenia, up to a minimum value of 8.45, with a co-morbid decrease in HBG (minimum value of 14.2). This time, discrete thrombocytopenia was noted, with a decrease in values over the 3 days from 202 to 146. Compared to the previous patient, this young man had low blood glucose values, with a MV of 100 mg/dL. The hepatocytolysis syndrome detected at the time of admission can be justified by MDMA ingestion, with GOT and TGP values exceeding the threshold of 1900 mg/dL. It should be noted that this patient had increased CK values (from 25,003 to 7383) (Table 5).

The evolution of the patients during the first 3 days post admission in the hospital was the following:

**Table 4 diagnostics-13-00940-t004:** Results of laboratory analysis patient L.I.

Patient L.I.	WBC	HBG	NEU	PLT	TP	INR	TMF	Glucose	TGP	TGO	Creatinine	RA	CK	Cl Serum	Sodium	CRP
23 April 2021	13.04	14.7	12.21	191	15.8	1.3		151	329	354	1.67	35		109.9	150	
24 April 2021																30.1
25 April 2021	14.85	13.2	12.84	130	15.6	1.28	Positive	169	243	155	1.16	34	3605	112.4	151	
26 April 2021	11.8	13	9.78	105	14.5	1.18	Positive	183	183	75	1.03	26	2671	112.1	148	

**Table 5 diagnostics-13-00940-t005:** Results of laboratory analysis patient C.I.

Patient C.I.	WBC	HBG	NEU	PLT	TP	INR	TMF	Glucose	TGP	TGO	Creatinine	RA	CK	CL Serum	Sodium	CRP
23 April 2021	12.26	15.06	10.33	202	14.1	1.14		87	2583	2583	1.14	25		103	141	
24 April 2021																
25 April 2021	8.55	14.8	6.35	160	15.1	1.24		120	1671	1701	0.9	26	25,003	104.8	139	
26 April 2021	8.45	14.2	5.88	146	14.6	1.19		93	309	480	0.81	25	7383	106.4	142	

The following pointsmay stand out:-The first patient, L.I., was taken in with extremely serious general condition, comatose, unresponsive to painful stimuli, GCS—3 points, required IOT + MV for stabilization. On arrival in the emergency room, the patient was fully monitored, but during consultations, presented with resuscitable CRA. The cardiological examination showed an EF of 40%, hypokinesis of the cardiac muscle, probably in the context of the existence of a post-internal IMA. During the other days of hospitalization, the parameters normalized, with a slowly favourable evolution.-The second patient, C.I., (with the same characteristics—age 32 years, male) showed the following at the time of pick-up by the ambulance crew: GCS = 15 points, conscious, hardly cooperative, responds with difficulty to painful and verbal stimuli. He associates liver damage during admission, with marked hepatocytolysis syndrome, with consequent increase in CK and LDH, respectively.

## 4. Discussion

The current study showed that intoxications among patients aged 17 years and over are frequently observed in the Emergency Department of the Galati Emergency Clinical Hospital. The most common intoxication was that of ethnobotanicals, which was present in the majority of males.

Vermes et al. conducted a similar descriptive study in the Erasmus M C University Medical Center in 2000, using a similar methodology. Comparison of the current study with the Vermes et al. study showed that the reported number of intoxications with alcohol, drugs or pharmaceuticals increased by 38% from 2000 to 2016. During this period, the population of Galați increased by 6.2% [28].

Our results on the prevalence of drug and pharmaceutical intoxications are in line with the literature, showing that approximately 0.3–2% of all Emergency Department visits are drug-related [1,2,7].

In the current study, ethnobotanicals, heroin, ecstasy and the amphetamine group—MDMA—were the most commonly observed. This can be explained by the current state policy of tolerance. The alarming increase observed in the current study is in line with the increasing trend observed in the general population in Romania for the use of cannabis, cocaine, MDMA, amphetamines and ethnobotanicals [29].

Rates of patients receiving therapy, with lower rates for patients receiving antidotes or gastric lavage, were lower compared to the rates found in the Dutch study by Ambrosius et al. (2012). A study among Dutch hospitals showed that the use of antidotes and gastric lavage varied among hospitals in the Netherlands and was mostly used in poisoning with pharmaceutical drugs. A relatively small group of these poisonings in the current study could be a possible explanation for the low rates found. Additionally, gastric lavage is performed only if the patient presents to the Emergency Department within the time frame in which it is merited and if the severity of the intoxication outweighs the potential risk of this intervention. In most cases of alcohol and ethnobotanical intoxication (which is by far the largest group in this study), there is no merit to this intervention. Additionally, for most of the intoxications observed in the current study, there is no suitable antidote.

Most patients were seen over the weekend. This is also frequently described in the literature [26,27,30,31,32,33,34]. In the current study, a quarter of all patients were admitted, which is comparable to rates in the literature, although rates varied widely from 25–78.3% [1,4,7,19,30].

Chemicals (chlorine, detergent, thinner, antifreeze, herbicides or organophosphorus) and other unknown substances accounted for 29% of all poisonings that received symptomatic treatment. A total of 47 patients were poisoned with substances other than chemical or pharmaceutical substances, alcohol and drugs.

By exposing these two cases, two possible developments of patients, known as users of psychoactive substances (MDMA), were brought to light.

Thus, the first patient was taken with extremely serious general condition, intubated and mechanically ventilated, hemodynamically stable, later developing cardio respiratory arrest. The patient regained vital functions after advanced resuscitation manoeuvres. Cardiac ultrasound revealed an EF of 40%, hypokinesis of the cardiac muscle, probably in the context of a post-internal acute infarction of the myocardium (IMA).

The second patient, C.I., was conscious but hardly cooperative on cardiac ultrasound without pathological changes, showing only sinus tachycardia. Both patients had associated liver damage during hospitalization, with marked hepatocytolysis syndrome, with consecutive increase in CK, respectively, LDH and mild thrombocytopenia.

It is important to mention the favourable but difficult evolution of the L.I. patient (with resuscitable CRA in the Emergency Department), most probably due to the patient’s clinical features, detected from the moment of his arrival in the Emergency Department.

The cardiovascular effects of amphetamines and their precursors have also been reported in the literature, being mainly related to the activation of the sympathetic nervous system through the release of norepinephrine, dopamine and serotonin. Circulating catecholamine concentrations can be increased up to fivefold [31].

Sympathetic activation can lead to varying degrees of tachycardia, vasoconstriction and significant variations in blood pressure, depending on the dose administered and the presence of pre-existing cardiovascular disease. Hypotension is causedby a relative state of catecholamine depletion, paradoxical central nervous system suppression (amphetamine) or acute myocardial depression (due to ischemia or direct toxic effect of the drug) [33].

Ischemia and myocardial infarction may be related to increased catecholamine concentration causing increased oxygen demand, coronary artery spasm, thrombocyte aggregation and thrombus formation [34].

Repetitive episodes of coronary artery spasm and paroxysms of hypertension can lead to endothelial damage, coronary artery dissections and accentuation of atherosclerosis. Paroxysmal increases in blood pressure can lead to aortic dissection or valvular damage, which increases the risk of endocarditis [35].

Adverse cardiovascular changes and sympathetic stimulation associated with these agents predispose to myocardial electrical instability and a wide range of tachyarrhythmias. The class 1 antiarrhythmic properties of cocaine may affect cardiac conduction, leading to conduction defects and bradyarrhythmia’s, including sinus arrest and atrioventricular block [36,37].

### Limits of the Study

A possible limitation of the current study was that most of the data presented were not obtained from toxicological analyses but from electronic medical records. Therefore, the figures presented could be an underestimation or overestimation of the actual occurrence of poisonings.

Additionally, for a minority of all transferred patients (24%; 6% of all patients), the length of stay in general hospitals was based on estimation, as we assumed that the length of stay in a general hospital ward was similar to the length of stay in the Galati Emergency Clinical Hospital.

For the two patients intoxicated with MDMA, it was not possible to identify the dose ingested by each patient in order to determine whether there was a direct link between the dose ingested and the patient’s evolution.

MDMA is a psychostimulant drug that displays effects related to amphetamine-type drugs plus a number of distinctive ones (closeness to others, facilitation to interpersonal relationship and empathy) that have been named by some authors as entactogen properties [37].

MDMA-induced acute toxic effects are related to its pharmacologic actions. Regarding neurotoxicity, only in the case of MDMA may a metabolic bioactivation be involved in long-term neurotoxic effects [37].

Genetic factors play a crucial role in drug metabolism with important implications in forensic toxicology. Gene variants of enzymes that metabolize or transport drugs change the bioavailability and the therapeutic/toxic dose of the drug. In forensic toxicology, whole blood sampling is preferred for the pharmacogenetic study [38].

Finally, it is important to note that this is a single-centre study that was conducted in Galati and, therefore, generalizability to other hospitals in Galati and beyond may be limited. We therefore recommend the creation of a national or preferably the international database to investigate regional differences in the occurrence, characteristics and health care costs of poisoning.

## 5. Conclusions

Poisoning among patients aged 17 years and over was commonly seen in the Emergency Departments of a large tertiary hospital in the city centre, accounting for 3.2% of all patients. In a third of the poisonings, more than one substance was used. Intoxication with ethnobotanicals was the most frequently observed, followed by use of drugs from the amphetamine group. The majority of patients presenting to the Emergency Department were male.

By exposing two cases, two possible outcomes of patients, known to be users of psychoactive substances, as well as cardio vascular effects due to MDMA consumption, were brought to light.

Therefore, this study suggests further research on hazardous alcohol consumption and drug abuse.

## Figures and Tables

**Figure 1 diagnostics-13-00940-f001:**
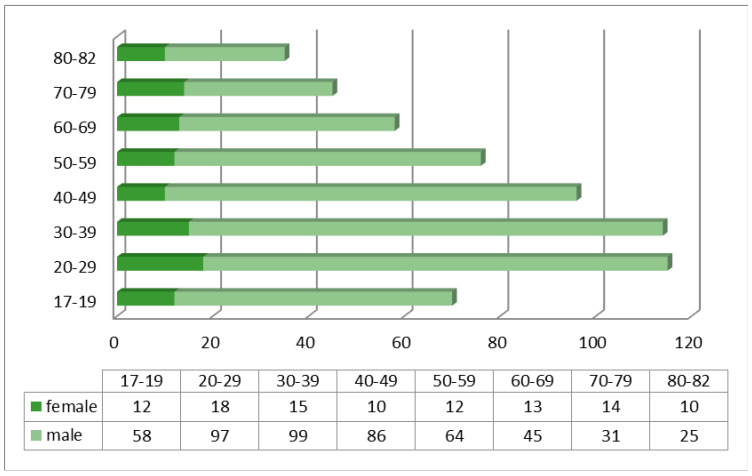
Distribution of patients with poisoning by gender and age.

**Figure 2 diagnostics-13-00940-f002:**
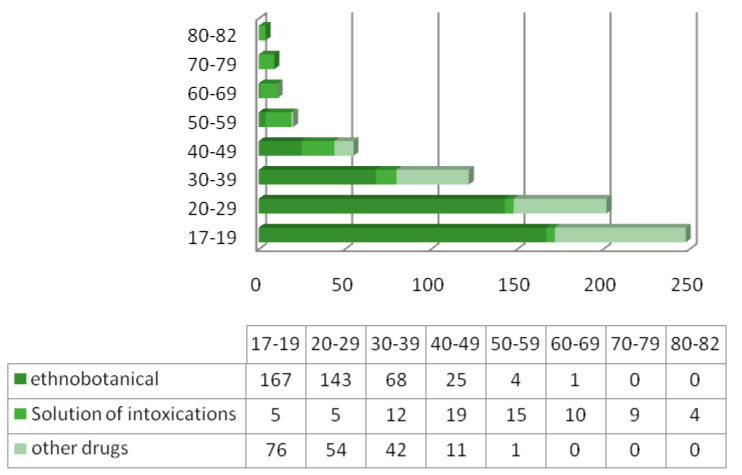
Distribution of poisonings by the age.

**Figure 3 diagnostics-13-00940-f003:**
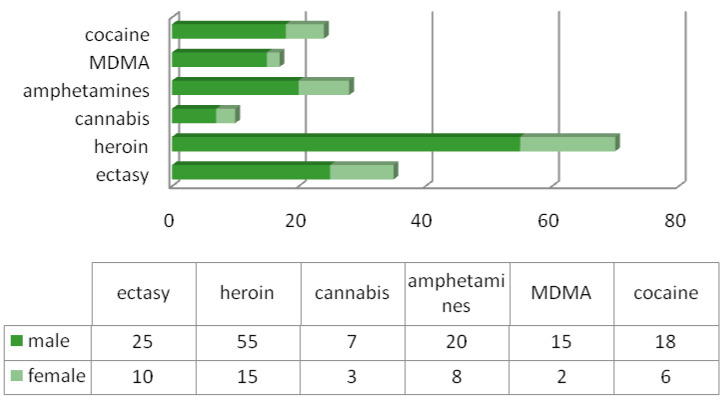
Distribution of drug intoxication by gender.

**Figure 4 diagnostics-13-00940-f004:**
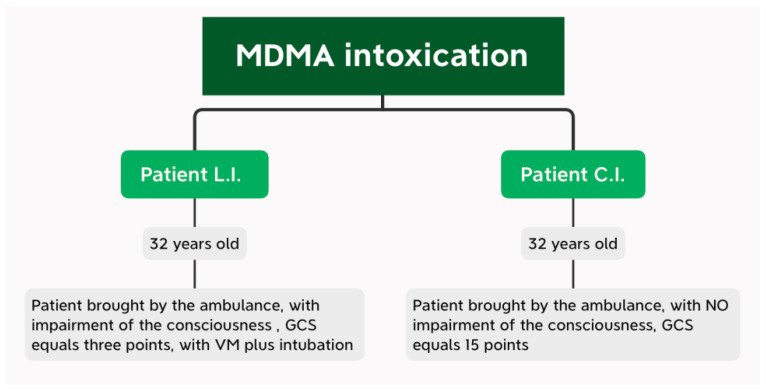
Primary examination of patients on admission to the Emergency Department.

**Table 1 diagnostics-13-00940-t001:** Number of patients consulted in the Emergency Department 2014–2016.

Year	Number of Patients
2014	86,560
2015	85,610
2016	85,990

**Table 2 diagnostics-13-00940-t002:** Patients distribution by type of intoxication and the day of the week which they presented in the Emergency Department.

Type of Intoxication	Day
Intentional exposure	85
Unintentional exposure	15
Day of presentation in ED	
Monday-Thursday	38
Friday-Sunday	62
Time of presentation	
Presentation hour 08:00–20:00	55
20:00–08:00	45
Admission to Internal Medicine Department	25
Intensive Care Admission	15
Discharge home	66
Addiction centre/police orientation	5
Left Emergency Department without medical advice	4

**Table 3 diagnostics-13-00940-t003:** Distribution of solution intoxications by gender.

Solution of Intoxications	Female	Male
Chlorine	8	9
Detergent	4	3
Paint	1	1
Hydrogen Peroxide	1	0
Rodenticide	2	3
Antifreeze	2	2
Furadan	1	1
Sodium hydroxide	1	3
Organophosphorus	3	6
Pesticide	1	0
Formaldehyde	1	0
Carbon-monoxide	0	3
Pre-release	0	1
Thinner	0	4
Descaling agent	3	1
Tomoxan	0	1
Royal water	0	1
Iodpovidone	0	1
Copper Sulphate	0	1
Petrol	1	0
Diesel	0	1
Plant fertiliser	0	1
Paint and thinner	0	1
Chlorine and detergent	0	3
Chlorine and antifreeze	0	1

## Data Availability

The data presented in this study are available on request from the corresponding author. The data are not publicly available due to privacy and ethical limitations.

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
