# Peer review of "Incidence of Intoxications in the Emergency Department of Galati Hospital with Examples of Cardiovascular Effects of MDMA Intoxication"

_diagnostics, 2023, doi:10.3390/diagnostics13050940_

Round 1

Reviewer 1 Report

This is an interesting paper dealing with the problem of illicit drug intoxication in Romania. The reviewer finds this paper important and interesting, yet has some minor comments to consider before he could recommend this work for publication. The reviewer does not like the form adopted, where there are very short paragraphs that break its continuity. Also, the introduction presented here should be considerably improved. The authors are missing a few points here.

1.) MDMA is considered an experimental therapeutic drug in the treatment of mental issues, i.e. depression, PTSD, or anxiety. It would be very beneficial if the authors could present ´a bright side´ of this drug, see e.g.  https://doi.org/10.12740/PP/OnlineFirst/133919 https://doi.org/10.1007/s00213-003-1705-6

2.) What is the motivation, and what is the profile of recreational drug users in Romania? A similar study regarding Poland has been presented, e.g. in https://doi.org/10.12740/PP/OnlineFirst/134317

3.) Referring to the above, it is unclear why this issue appears in Romania. Is that a self-healing motivation or mainly stem from a vibrant nightclub scene in Romania?

4.) The authors use the term MDMA, whereas, it is very unlikely that the street drugs in Romania contain the dominant amount of MDMA but other psychoactive substances like MDA, which is more psychedelic in action and easier to synthesize ¨Recent changes in Europe’s MDMA/ecstasy market´ TD0116348ENN.pdf (europa.eu)

The authors should also comment on the possible origin of these drugs on the Romanian street market. Are these drugs produced in Romanian illegal labs or are smuggled from the west? Any information about the chemical profiles? Common contaminations? Are there any synthetic routes identified by Police?

The reviewer is aware that this is perhaps a pioneering report on MDMA use in Romania, yet the lack of any sociologic context reduces the quality of this work as in the end it mainly shows the statistics.

The reviewer support publication of this work, yet after minor revision addressing these questions. The overall quality of this paper is good and the topic is certainly interesting. 

Author Response

This is an interesting paper dealing with the problem of illicit drug intoxication in Romania. The reviewer finds this paper important and interesting, yet has some minor comments to consider before he could recommend this work for publication. The reviewer does not like the form adopted, where there are very short paragraphs that break its continuity. Also, the introduction presented here should be considerably improved. The authors are missing a few points here.

1.) MDMA is considered an experimental therapeutic drug in the treatment of mental issues, i.e. depression, PTSD, or anxiety. It would be very beneficial if the authors could present ´a bright side´ of this drug, see e.g. https://doi.org/10.12740/PP/OnlineFirst/133919 https://doi.org/10.1007/s00213-003-1705-6
- MDMA was first used in the 1970s as an aid in psychotherapy (treatment of mental disorders using "talk therapy"). The drug did not have the support of clinical trials (human studies) or approval from the U.S. Food and Drug Administration (FDA). In 1985, the U.S. Drug Enforcement Administration (DEA) labeled MDMA as an illegal drug without recognized medicinal use. However, some researchers remain interested in its value in psychotherapy when given to patients under carefully controlled conditions. MDMA is currently in clinical trials as a possible treatment aid for post-traumatic stress disorder (PTSD); for anxiety in terminally ill patients; and for social anxiety in adults with autism. Recently, the FDA gave MDMA-assisted psychotherapy for PTSD a groundbreaking therapy designation.
In Romania MDMA is not used for therapeutic purposes, so there are no statistical studies available to confirm or deny the beneficial effects of this drug in some pathologies.

2.) What is the motivation, and what is the profile of recreational drug users in Romania? A similar study regarding Poland has been presented, e.g. in https://doi.org/10.12740/PP/OnlineFirst/134317
The causes of drug use are numerous and depend on a number of factors, the most important of which are socio-cultural factors (belonging to a group where drugs are used, seeking a way to protest, social isolation, non-integration, living exclusively in the present, etc.) and individual factors (intolerance of frustration, the need for satisfaction, pathological aggression, maladjustment which can lead to deviant behaviour, extremely early disrupted parent-child relationships, psychopathological disorders linked to adolescent crises, psychopaths, etc.). It is also extremely important to bear in mind that the pressure of the social group to which a person belongs and the availability and ease of obtaining drugs are determining factors in the initiation and maintenance of drug use.

3.) Referring to the above, it is unclear why this issue appears in Romania. Is that a self-healing motivation or mainly stem from a vibrant nightclub scene in Romania?
According to the most recent survey in the general population (15-64 years), GPS 2019, for the first time, new psychoactive substances (NPS) - 6.3% - rank first in the order of most consumed illicit drugs in Romania. Next: cannabis - 6.1%, cocaine/crack -1.6%, non-prescription drugs - 1.5%, ecstasy - 1.0%, heroin - 0.9%, LSD - 0.5%, amphetamines - 0.2%, solvents/inhalants - 0.1%.
At the same time, according to the most recent survey in the school population, ESPAD 2019 (16 years), the most consumed illicit drug among adolescents continues to be cannabis/hashish - 8.7%. This is followed by NSP at 3.2%, solvents/inhalants at 2.8%, cocaine at 1.8%, LSD or other hallucinogens - 1.7%, ecstasy - 1.2%, heroin - 0.7%, ketamine - 1.8%, crack - 0.6%, methamphetamines - 0.6%, amphetamines -0.5% and GHB - 0.4%.
http://ana.gov.ro/wp-content/uploads/2021/01/RN_2020_final.pdf.
The study "Drug use among young people who frequent recreational environments", carried out by the National Anti-Drug Agency (ANA) aimed to explore the problem of drug use among the young population in Romania, aged between 15 and 34 years, who frequent different leisure venues (festivals, concerts, clubs, etc.), in order to develop intervention programs tailored to the specifics of this segment of the population.
The study was conducted on a nationally representative sample of 7200 respondents.
non-institutionalised population, aged 15-64 years, at the time of the survey.
1 January 2019. Within the sample, oversampling was used on
Bucharest-Ilfov region, of 1500 subjects for people aged 15-34. Data collection took place between September and November 2021, and data analysis and interpretation were carried out in 2022. The results of the study were as follows:

- in recreational settings, the extent of lifetime use of any illicit drug among young people aged 15-34 is higher than that observed in the General Population Study (GPS 2019), for the population segment 15-34 years (GPS 2019 -11%, compared to 21.8% - online, respectively 18.5% - festivals/concerts)
- Prevalences of use for most of the drugs studied
values are at least twice as high as those
observed in the General Population Survey (GPS 2019) for
population segment 15-34 years:
- Cannabis: GPS 2019 - 6%, Study in recreational settings -20.3% - online,
compared to 17.3% - face to face
- Cocaine: GPS 2019 - 0.7%, Recreational survey -4.7% - face to face
to face, compared to 2.8% - online
- Ecstasy: GPS 2019 - 0.8%, Recreational survey - 6.5% - face to face
face, compared to 4,4% - online
- NPS: GPS (General Survey Population) 2019 - 5.1%, Recreational survey - 4.1% - face to face, vs. 2.6% - online

4.) The authors use the term MDMA, whereas, it is very unlikely that the street drugs in Romania contain the dominant amount of MDMA but other psychoactive substances like MDA, which is more psychedelic in action and easier to synthesize ¨Recent changes in Europe’s MDMA/ecstasy market´ TD0116348ENN.pdf (europa.eu)
The authors should also comment on the possible origin of these drugs on the Romanian street market. Are these drugs produced in Romanian illegal labs or are smuggled from the west? Any information about the chemical profiles? Common contaminations? Are there any synthetic routes identified by Police?
The origin of drugs in Romania is somewhat recognized, in terms of the routes used by organised crime networks. Thus, heroin is trafficked along the route Afghanistan - Pakistan - Iran - Turkey - Greece - former Yugoslav states and Western European countries, with Romania, England and the Netherlands as destinations. Cocaine comes from Colombia, Bolivia, Peru and Venezuela and generally follows the route Spain - France - Austria - Hungary to Romania or South America - West and Central Africa - Romania - to Central and Western Europe. This type of drug is mainly found in big cities - Bucharest, Timisoara, Constanta - being a high-priced drug, a luxury drug, expensive for the economic level of most users in Romania. Cannabis comes from Spain, Greece, Bulgaria, the Netherlands or Albania transiting Serbia or Bulgaria. The significant increase in domestic cannabis cultivation reflects a shift in the focus of traffickers' activities to avoid the risks posed by potential international shipments.
Synthetic drugs have experienced a particular expansion on the Romanian market in the last two years, taking into account the fact that, in addition to classic substances, derivatives known as "designer drugs" have also appeared.
These new substances with psychoactive properties have been improperly called ethnobotanicals or legal substances.
The presence of these drugs on the Romanian market has also been driven by the free movement of people and goods, but also by price differences between synthetic drugs sold on the Western European market and those sold on the Romanian market. In Romania, there have also been attempts to set up laboratories to produce these substances.
Synthetic drugs (amphetamines, methamphetamines, ecstasy) continue to come from Western European countries, in particular the Netherlands, and are transported either by courier, air or land.
ANA (national anti-drug agency)
MDMA, as an illicitly sold drug in Europe, comes in the form of professional-looking tablets stamped with different symbols, depending on the manufacturer. However, the composition of the tablets varies greatly, both in terms of the drugs they contain and the quantities. Several different laboratories have analysed samples sold on the street and found that the drug sold as "ecstasy" can be MDMA, MDEA, MDA, PMA (para-methoxyamphetamine), ephedrine or various mixtures of these. The typical dosage range of MDMA for recreational use varies from 50 mg to 150 mg [1,2], but the amount per tablet in different batches of tablets can vary 70 times or more, from almost zero to well over 100 mg.

The reviewer is aware that this is perhaps a pioneering report on MDMA use in Romania, yet the lack of any sociologic context reduces the quality of this work as in the end it mainly shows the statistics.
The reviewer support publication of this work, yet after minor revision addressing these questions. The overall quality of this paper is good and the topic is certainly interesting.

Reviewer 2 Report

The manuscript presents a study on the frequency of intoxication in an emergency unit in Romania, referring also to the nature of the substances that caused it. Although it includes an appreciable number of patients (609 patients), I could not say that the study is very complex. I would recommend the authors, as far as possible, to improve the study.

In addition, there are some spelling mistakes or wrong expressions that I have marked in the text. I would like to suggest some points for revision:

page 2 - MDMA is not structurally similar to serotonin

page 6 - Fig 2, please write in English; 408 patients vs ethnobotanicals; Fig 3, please write in English

page 8 - Please explain the abbreviation VM in Fig 4

page 10 - Since not all abbreviations are explained at the end of the manuscript, please explain the significance for PLT, CRP, MV, TGO, TGP, CK at the first appearance in the text, as well as the abbreviations in Tables 4 and 5 for a better understanding of the readers.

page 14 - references in English.

Author Response

The manuscript presents a study on the frequency of intoxication in an emergency unit in Romania, referring also to the nature of the substances that caused it. Although it includes an appreciable number of patients (609 patients), I could not say that the study is very complex. I would recommend the authors, as far as possible, to improve the study.
In addition, there are some spelling mistakes or wrong expressions that I have marked in the text. I would like to suggest some points for revision:

page 2 - MDMA is not structurally similar to serotonin
Ecstasy' (MDMA) and related drugs are amphetamine derivatives that also have some of the pharmacological properties of mescaline. Ecstasy (MDMA) is the popular name for a substance chemically identified as N-methyl-3,4-methylenedioxyamphetamine or 3,4-methylenedioxy-methamphetamine. A closely related compound is N-ethyl-3,4-methylenedioxyamphetamine. The name 'ecstasy' is used somewhat indiscriminately. In earlier years, the name was also given to 3,4-methylenedioxyamphetamine (MDA). The three compounds are closely similar in their chemistry and biological effects. Ecstasy differs from amphetamine and methamphetamine, but resembles the structure of the hallucinogenic material mescaline. As a result, the pharmacological effects of MDMA are a mixture of those of amphetamines and mescaline.
It's not serotonin, it's mescaline (that was a transcription error, probably)

page 6 - Fig 2, please write in English; 408 patients vs ethnobotanicals; Fig 3, please write in English
Done

page 8 - Please explain the abbreviation VM in Fig 4
Done VM mechanical ventilation

page 10 - Since not all abbreviations are explained at the end of the manuscript, please explain the significance for PLT, CRP, MV, TGO, TGP, CK at the first appearance in the text, as well as the abbreviations in Tables 4 and 5 for a better understanding of the readers.
Done

page 14 - references in English.
Done

Round 2

Reviewer 2 Report

Please pay attention to the text marked in yellow.

Author Response

Dear Reviewer,

The final reviewed form of the manuscript.

Thank you for your collaboration.
